

# Chimpanzees (*Pan troglodytes*) display limited behavioural flexibility when faced with a changing foraging task requiring tool use

Rachel A. Harrison[1,2] and Andrew Whiten[1]

[1] Centre for Social Learning and Cognitive Evolution, School of Psychology & Neuroscience, University of St Andrews, St Andrews, United Kingdom
[2] School of Psychology, University of Birmingham, Birmingham, United Kingdom

## ABSTRACT

Behavioural flexibility, the ability to alter behaviour in response to environmental feedback, and to relinquish previously successful solutions to problems, is a crucial ability in allowing organisms to adapt to novel environments and environmental change; it is essential to cumulative cultural change. To explore this ability in chimpanzees, 18 individuals (*Pan troglodytes*) were presented with an artificial foraging task consisting of a tube partially filled with juice that could be reached by hand or retrieved using tool materials to hand. Effective solutions were then restricted in the second phase of the study by narrowing the diameter of the tube, necessitating the abandonment of previously successful solutions. Chimpanzees showed limited behavioural flexibility in comparison to some previous studies, increasing their use of effective techniques, but also continuing to attempt solutions that had been rendered ineffective. This adds to a literature reporting divergent evidence for flexibility (the ability to alter behaviour in response to environmental feedback, and to relinquish previously successful solutions to problems) versus conservatism (a reluctance or inability to explore or adopt novel solutions to problems when a solution is already known) in apes.

## INTRODUCTION

Behavioural flexibility, the ability to alter behaviour based upon environmental feedback and to inhibit previously successful behaviours, is an ability that allows organisms to adapt their behaviour to suit changing or novel environments and supports problem solving by allowing individuals to adapt their behaviour to success or failure at a problem (*Sol, Timmermans & Lefebvre, 2002*; *Griffin & Guez, 2014*; *Chow, Lea & Leaver, 2016*; *Audet & Lefebvre, 2017*). Behavioural flexibility can also describe the capacity for, and interest in, continuing to acquire novel solutions to an unchanging problem for which a solution is already known (*Lehner, Burkart & Van Schaik, 2011*), though most experimental explorations of behavioural flexibility incorporate changes in task parameters (and therefore changes in environmental feedback). Whilst innovation (defined by *Reader & Laland 2003*, pp.14, as "A process that results in new or modified learned behaviour

Corresponding author
Andrew Whiten, aw2@st-andrews.ac.uk

and that introduces novel behavioural variants into a population's repertoire'') has been suggested to be a component of behavioural flexibility (*Lehner, Burkart & Van Schaik, 2011*), we also consider the acquisition of novel behaviours via social learning to constitute flexible behaviour (*Wright et al., 2010*), and indeed the application of known behaviours to a novel problem. Behavioural flexibility is thought to be a key ability supporting the evolution of cumulative culture (*Dean et al., 2014*). With culture defined as "group-typical behaviour patterns shared by members of a community that rely on socially learned and transmitted information" (*Laland & Hoppitt, 2003*, pp.151), cumulative culture is the process whereby these socially learnt behaviours are modified and the modifications are retained, resulting in behaviours and technologies more complex than an individual could invent within their lifetime (*Tomasello, 1994*; *Tennie, Call & Tomasello, 2009*). As cumulative culture relies upon the modification of known behaviours it necessitates flexibility in both the process of innovation by some individuals, and in the acquisition by others of the improved behaviours that result.

**Behavioural flexibility and cumulative culture**

Evidence for cumulative culture in our closest living relative, the chimpanzee, is limited and controversial (*Dean et al., 2014*; *Whiten, 2017*), with *Boesch (2003)* highlighting three behaviours observed in wild chimpanzees (nut-cracking incorporating additional stones to stabilise the anvil, parasite manipulation in which parasites are placed on a leaf which is then folded and cut, and well-digging incorporating the use of leaf-sponges to retrieve water from deep wells) as potential evidence of cumulative culture, whilst *Sanz, Call & Morgan (2009)* describe apparent improvements made to termite fishing tools by chimpanzees in the Goualougo Triangle, Republic of Congo. However, such interpretations require assumptions to be made regarding the social transmission of these behaviours, whether they represent greater complexity than that achievable by an individual alone, and indeed whether they are the result of cumulative progressions rather than unconnected innovations (which might be better described as 'accumulation'; *Dean et al., 2014*). With such limited evidence of cumulative culture in chimpanzees, investigation of the abilities required to support cumulative culture is required. We suggest that behavioural flexibility, along with innovation and social learning, should be considered and investigated as a potential limiting factor for cumulative culture. These three capacities are expected to work in concert to support cumulative culture—an individual innovates an improvement to a behaviour or tradition, and this improvement is passed on via social learning to other group members. As cumulative culture requires the modification of known behaviours, flexibility is required in both the innovator, to modify a behaviour within their repertoire, and in the group members socially acquiring this modified behaviour (that, in a process of cumulation, is expected to replace a known behaviour which previously served the same purpose). As outlined by *Charbonneau (2015)*, an ability to innovate entirely novel behaviours 'from scratch' alone is not sufficient to support cumulative culture, which instead requires an ability to modify *known* behaviours. High-fidelity social transmission is believed to be critical in supporting cumulative culture, as it prevents backwards 'slippage' or the loss of modifications to behaviours (*Tennie, Call & Tomasello, 2009*; *Lewis & Laland, 2012*).

However, somewhat paradoxically, at least some individuals in a population must also be capable of modifying behaviours, having acquired them via high-fidelity social learning.

## Measuring behavioural flexibility

Behavioural flexibility has been assessed at a species level via the proxy measure of innovation frequency (for example, *Lefebvre et al., 1997*; *Lefebvre, Reader & Sol, 2004*, for avian comparisons), and innovation has been considered a component of behavioural flexibility (*Lehner, Burkart & Van Schaik, 2011*; *Audet & Lefebvre, 2017*). This makes the assumption that behavioural flexibility leads to increased innovativeness, which may not always be the case at either an individual or species level (*Griffin et al., 2013*; *Logan, 2016*). Behavioural flexibility in response to environmental change has been tested directly in a range of species, both in the wild and captivity, frequently by employing experimental paradigms in which animals must respond to a change in task parameters, such as reversal learning (*Bond, Kamil & Balda, 2007*; *Boogert, Monceau & Lefebvre, 2010*; *Manrique & Call, 2015*; *Liu et al., 2016*) or multi-access puzzle box tests (*Auersperg et al., 2011*; *Lehner, Burkart & Van Schaik, 2011*; *Manrique, Völter & Call, 2013*; *Richter, Hochner & Kuba, 2016*).

Both intra- and interspecific differences in flexibility have been found using these experimental techniques. Individual differences in behavioural flexibility within species have been shown in animals as diverse as octopuses, pigs and mice (*Benus et al., 1990*; *Richter, Hochner & Kuba, 2016*; *Bolhuis et al., 2004*), as well as in wild chimpanzees and sanctuary-housed orangutans (*Gruber, 2016*). Such inter-individual differences within species are perhaps not surprising, given potential differences between individuals in terms of task motivation, cognitive ability, and personality (the latter having been shown to impact performance on problem solving tasks in chimpanzees; *Massen et al., 2013*). In the context of cumulative culture, an understanding of which individuals are most capable of behavioural flexibility, and investigation of any other traits that might co-occur with behavioural flexibility on an individual level, may allow predictions to be made regarding which individuals or demographics are likely to contribute to the modification of behaviours in a population's repertoire.

In addition, interspecific differences in behavioural flexibility have been demonstrated in both corvids and great apes (*Bond, Kamil & Balda, 2007*; *Manrique, Völter & Call, 2013*), and investigation of these interspecific differences allows exploration of hypotheses regarding the evolution of behavioural flexibility. *Bond, Kamil & Balda (2007)* argue that behavioural flexibility may be more apparent in species with highly complex social systems, in order to cope with rapidly fluctuating social contexts, and found that pinyon jays, a highly social species, had lower error rates on a serial reversal learning tasks than Clark's nutcrackers (a relatively solitary species with specialised spatial memory supporting caching behaviour) and western (California) scrub jays (a generalist species in terms of both ecology and social behaviour). Continued investigation of species' abilities to behave flexibly will allow investigation of the evolutionary pressures that lead to high behavioural flexibility, and will also allow investigation of other cognitive abilities that may co-evolve with behavioural flexibility. For example, it has also been shown that pinyon jays outperform

Clark's nutcrackers on social learning tasks (*Templeton, Kamil & Balda, 1999*). Given that a suite of abilities is likely required to support cumulative culture (*Dean et al., 2014*), an understanding of why component abilities may emerge, and whether they share common evolutionary origins, is likely to enhance our understanding of the evolution of cumulative culture.

Observation of the tool preferences of wild tool-using species also provides some indication of the flexibility of such species. Examination of the tool preferences of wild New Caledonian crows found strong, persistent local preferences in terms of the plant species from which their hook-tools are manufactured (*St Clair et al., 2016*), despite access to alternative plants, which are preferred, at another, nearby study site. This may indicate some level of individual conservatism (a reluctance to explore alternative solutions) in this species' tool manufacture, though the authors also point out that it shows that crows at one study site have either switched from a previous tool material, or acquired hooked stick tool-use relatively recently. Both scenarios would reflect flexibility, both on the part of initial innovators and any individuals that may have socially acquired this alternative behaviour. Study of individual crows' tool manufacture indicates multiple bending techniques and adjustments during manufacture, which would seem to indicate flexibility (*Klump et al., 2015*; *Rutz et al., 2016*). Wild chimpanzees have been observed to modify their tool-use behaviours, with an individual in Bossou being observed firstly applying a tool and technique generally used for ant-dipping on the ground to ant-fishing in trees, and later using tools of a length better suited to ant-fishing in trees (*Yamamoto et al., 2008*). Wild chimpanzees have also been observed to socially learn novel tool behaviours, such as crafting sponges from moss rather than leaves (*Hobaiter et al., 2014*). These observations of wild behaviour provide some insight into chimpanzees' capacity for behavioural flexibility.

## Chimpanzees—behaviourally flexible or conservative?

In several studies, chimpanzees have been reported to be conservative, rather than flexible, in their approach to artificial foraging tasks, continuing to use a habitual solution despite the prospect of gaining a greater reward via behavioural change (*Marshall-Pescini & Whiten, 2008*) or the habitual solution being made impossible or unrewarding (*Hrubesch, Preuschoft & Schaik, 2009*; *Bonnie et al., 2012*). Behavioural conservatism has also been reported in wild chimpanzees, with *Gruber et al. (2009)* finding that two different communities relied upon their respective habitual tool behaviours when faced with a novel artificial honey-dipping task, in one community failing to acquire the useful tool behaviour used by the other community even when scaffolded towards it (*Gruber et al., 2011*; *Gruber, 2016*). Similarly, *Cardoso & Ottoni (2016)* found that providing two communities of wild bearded capuchin monkeys with a dipping task resulted in only the group that already habitually used probing tools solving the task, again despite efforts to scaffold the non-probe-using group towards this behaviour. Chimpanzees may also continue to perform behaviours when these behaviours are no longer necessary, for example, continuing to avoid a non-functional trap in the inverted trap-tube problem (*Povinelli, 2000*; though see also *Mulcahy & Call, 2006*), although it should be noted that in paradigms in which continuing this behaviour is equally rewarding as ceasing to avoid the trap, adult humans also continue to avoid non-functional

traps (*Silva, Page & Silva, 2005*). Related to these findings of apparent conservatism or lack of flexibility in chimpanzees is the concept of 'functional fixedness': the inability to invent a novel use for a tool with which the animal already has experience (*Hanus et al., 2011*; *Brosnan & Hopper, 2014*). *Hanus et al. (2011)* found that captive chimpanzees were more likely to solve the 'floating peanut' experiment (in which water must be added to a tube in order to raise a floating peanut to a level at which it can be reached) when a novel water dispenser was added to their enclosure. The authors suggest that the old dispenser had a fixed function for the animals, which prevented them from discovering it as a potential task solution.

By contrast, other recent studies have implied that under certain conditions, great apes may be capable of flexibly altering their behaviour in response to a changing task (*Lehner, Burkart & Van Schaik, 2011*; *Manrique, Völter & Call, 2013*; *Yamamoto et al., 2008*; *Davis et al., 2016*; *Vale et al., 2017*). In the studies of both *Lehner, Burkart & Van Schaik (2011)*, and *Manrique, Völter & Call (2013)*, in order to successfully retrieve a reward, great apes had to not only flexibly alter their behaviour in response to changing tasks, but also develop novel methods of solving the foraging tasks that they were presented with. *Lehner, Burkart & Van Schaik (2011)* found that orangutans were capable of inventing novel solutions to a task in which juice could be retrieved from a tube by dipping tool materials into the tube. When the width of the tube was narrowed, orangutans flexibly altered their behaviour, abandoning previously successful solutions and inventing novel solutions, some of which the authors argue are cumulative improvements upon previous solutions. It should be noted that although novel solutions were not experimentally seeded in the *Lehner, Burkart & Van Schaik* study (*2011*), orangutans were tested under group conditions, and so potentially had access to social information. In a study testing all four nonhuman species of great ape, in which subjects were tested individually (eliminating the possibility of social learning), *Manrique, Völter & Call (2013)* found that all species were able to abandon previous solutions and invent novel solutions to a non-tool-based artificial foraging task in which solutions were rendered obsolete in three stages, though orangutans performed more poorly than other species, only solving two stages. These studies demonstrate that great apes can be capable of flexibly altering their behaviour and relinquishing previously successful behaviours in order to solve changing artificial foraging tasks, without the benefit of experimentally-seeded solutions.

Studies have also shown that chimpanzees are able to socially acquire more efficient or more rewarding solutions to problems, having already mastered a less efficient solution. *Yamamoto, Humle & Tanaka (2013)* found that chimpanzees provided with a task in which juice could be retrieved from within a tube via a small hole were capable of swapping from an inefficient 'dipping' technique to a more efficient 'straw-sucking' technique following observation of a conspecific employing the more efficient technique, while *Vale et al. (2017)* found that following the removal of simple tools that could be used to acquire juice in a dipping task, chimpanzees began to use complex tools that required modification (unscrewing a valve so a tube could be used as a 'straw' to suck up juice). *Davis et al. (2016)* found that chimpanzees were able to relinquish a highly inefficient task solution in favour of a more efficient solution when provided with demonstrations of the efficient solution

by a conspecific or human demonstrator. The authors of the latter study argue that such flexibility may be more apparent in situations in which there is a large difference in efficiency between two technique options (with a relatively highly inefficient solution more likely to be relinquished). This may suggest chimpanzees use a 'copy-if-dissatisfied' strategy when given the opportunity to socially acquire more efficient solutions to problems (*Laland, 2004*). If so, behavioural flexibility in such studies may be apparent only when the original solution is rendered highly inefficient and unsatisfying. The findings of *Lehner, Burkart & Van Schaik (2011)* and *Manrique, Völter & Call (2013)* may, in light of this, imply that great apes also employ an 'explore-if-dissatisfied' strategy when given the opportunity to achieve improved solutions to problems without the benefit of an existing competent model.

## STUDY AIMS

We aimed to investigate the capability of chimpanzees to alter their behaviour in response to an artificial foraging task in which viable solutions became restricted as time progressed, using a liquid-retrieval task comparable to that used by *Lehner, Burkart & Van Schaik (2011)* with orangutans.

### Behavioural flexibility

Chimpanzees were presented with the task as a group, and therefore had access to both individual and social information about the task and potential solutions. This means that individuals were not limited to what they themselves could invent in terms of task solutions, which we believe provides a more ecologically valid measure of behavioural flexibility, as this has been defined in the past as the continued acquisition of new solutions through either innovation or social learning (*Lehner, Burkart & Van Schaik, 2011*). If chimpanzees are able to respond flexibly to changes in task conditions (as shown by *Manrique, Völter & Call, 2013*), we would predict individuals to increase their use of techniques that remain effective, and to decrease their use of techniques that have been rendered ineffective, in the face of task restrictions. Alternatively, if chimpanzees are behaviourally inflexible, we predict the continued use of ineffective techniques and no increase in the use of effective techniques. If chimpanzees follow an 'explore if dissatisfied' strategy, we would predict the emergence of novel techniques following the imposition of task restrictions, whilst a 'copy if dissatisfied' strategy would result in the acquisition of effective techniques by individuals other than the technique's innovator following the imposition of task restrictions.

### Subgroups and the impact of developmental conditions

Chimpanzees were studied in Edinburgh Zoo, which houses a single community that originates from two separate subgroups; long-term residents of the zoo ('Edinburgh' chimpanzees), and others introduced in 2010 ('Beekse Bergen' chimpanzees). These subgroups are now well integrated socially (*Schel et al., 2012*; *Watson et al., 2015*) but have differing life histories, with the Beekse Bergen subgroup coming from a laboratory and first living in a zoo environment from 2007. The majority of chimpanzees in the Beekse Bergen subgroup were hand-reared in nurseries rather than mother-reared, whilst all captive-born chimpanzees in the Edinburgh subgroup were mother-reared. Both

rearing history and housing have been found to impact problem solving ability (and other behaviour) in chimpanzees (*Brent, Bloomsmith & Fisher, 1995*; *Bloomsmith et al., 2006*; *Morimura & Mori, 2010*; *Vlamings, Hare & Call, 2010*), so this study also considers the impact of subgroup membership upon task performance. Members of the Beekse Bergen subgroup, with a history of laboratory-housing and including a number of hand-reared chimpanzees, were predicted to perform more poorly on the task than long-term zoo residents.

## Scaffolding towards an effective solution

Learning through exposure to the artefacts of others' tool use is hypothesised to aid in the development of tool behaviour in wild populations (*Tennie, Call & Tomasello, 2009*); however, previous experimental studies of chimpanzees (*Gruber et al., 2011*) and capuchin monkeys (*Cardoso & Ottoni, 2016*) have not shown that providing scaffolding intended to approximate these artefacts (tool materials already inserted into a task, for example) leads to the acquisition of novel tool behaviours. By providing chimpanzees with limited exposure to scaffolding towards a novel tool technique in the final phase of our study, we examined whether chimpanzees could acquire novel tool behaviour through exposure to favourable affordances in the form of an effective tool material correctly inserted into the task, as might occur in the wild where proficient tool-users leave tools in place that novices may discover.

# METHODS

## Ethical approval

The study received ethical approval from the University of St Andrews Animal Welfare and Ethics Committee, and was approved by the Budongo Trail Research Committee. Research was conducted in accordance with the guidelines of the Association for the Study of Animal Behaviour.

## Subjects and study site

Subjects were 18 chimpanzees housed as one group in the Budongo Trail facility at Edinburgh Zoo (see Table 1 for details). The group is composed of two 'subgroups', long-term residents of the zoo ('Edinburgh' chimpanzees), and others introduced in 2010 ('Beekse Bergen' chimpanzees), each numbering nine individuals. The chimpanzees lived in three interconnecting indoor enclosures measuring 120 m$^2$ each and one outdoor enclosure measuring 1,832 m$^2$, along with two research 'pods' (two connected rooms measuring 26.5 m$^2$ in total) provided for cognitive research. Chimpanzees were fed six to eight scatter feeds throughout the day at random times and locations, with water available *ad libitum* throughout the facility. The research pods were accessible via overhead tunnels from the chimpanzees' indoor enclosures, and activity within the pods could not be viewed by chimpanzees outside the pods. Our task was presented in one of these research pods, and chimpanzees had access to both pods during testing. Chimpanzees in the second research pod during testing had limited visual access to the task through an open slide door separating the pods, while individuals in the primary pod (in which the task was presented)

**Table 1 Demographic and rearing information of the chimpanzees with level of participation in current study.**

| Subgroup | Individual | Sex | Year of birth (age at time of testing) | Origin (wild or captive born) | Rearing | Participation in current study (number of attempts) | | |
|---|---|---|---|---|---|---|---|---|
| | | | | | | Wide tube | Narrow tube | Narrow scaffolded |
| Edinburgh | Qafzeh | M | 1992 (22) | Captive | Mother | 0 | 1 | 0 |
| | Kindia | M | 1997 (17) | Captive | Mother | 14 | 4 | 1 |
| | Liberius | M | 1999 (15) | Captive | Mother | 1 | 0 | 1 |
| | David | M | 1975 (39) | Captive | Mother | 4 | 0 | 0 |
| | Louis | M | 1976 (38) | Wild | | 6 | 15 | 0 |
| | Lucy | F | 1976 (38) | Captive | Mother | 11 | 2 | 0 |
| | Kilimi | F | 1993 (21) | Captive | Mother | 175 | 103 | 87 |
| | Cindy | F | 1964 (50) | Wild | | 0 | 0 | 10 |
| | Emma | F | 1981 (33) | Captive | Mother | 37 | 59 | 0 |
| Beekse Bergen | Paul | M | 1993 (21) | Captive | Hand-raised | 0 | 0 | 0 |
| | Pearl | F | 1969 (45) | Wild | | 311 | 214 | 40 |
| | Sofie | F | 1981 (33) | Captive | Hand-raised | 3 | 0 | 2 |
| | Lianne | F | 1989 (25) | Captive | Mother | 0 | 0 | 0 |
| | Heleen | F | 1991 (23) | Captive | Mother | 4 | 0 | 0 |
| | Edith | F | 1996 (18) | Captive | Mother | 385 | 741 | 73 |
| | Eva | F | 1980 (34) | Captive | Hand-raised | 311 | 21 | 4 |
| | Frek | M | 1993 (21) | Captive | Hand-raised | 125 | 80 | 11 |
| | Rene | M | 1993 (21) | Captive | Hand-raised | 152 | 11 | 3 |

**Notes.**
In addition to their own attempts, all individuals with the exception of Cindy and Liberius were present in the research pods during another individual's attempt on at least one occasion.

generally had good visual access to the task. A maximum of seven chimpanzees were recorded as present in the primary pod during testing, with space for further individuals in the second pod. All chimpanzees had previous experience with artificial foraging tasks involving tool use, having been provided with the 'Panpipes' task (in which a stick tool is used to retrieve a grape from within the apparatus, see *Whiten, Horner & De Waal, 2005*, for details of the apparatus) and a food-raking task in which plastic stick tools could be used to retrieve out-of-reach rewards (see *Price et al., 2009*, for a similar task) (V West, 2013, unpublished master's thesis; R Harrison, 2013, unpublished master's thesis). In addition, chimpanzees had previous experience using a touchscreen device in the research pods (*Wallace et al., 2017*).

## Apparatus

Echoing the study of *Lehner, Burkart & Van Schaik (2011)*, we provided the chimpanzees with an artificial foraging task in which dilute Ribena juice could be retrieved from within transparent polycarbonate tubes, using either provided tool materials or hands. The task had two stages, for which two widths of tube were provided. The first ('wide') tube measured 30 cm tall with a 10 cm inner diameter (see Fig. 1), and the second ('narrow') tube measured 30 cm tall with a 5 cm inner diameter. These tubes were presented in the research pods, bolted to a polycarbonate screen facing into the pod, and could be

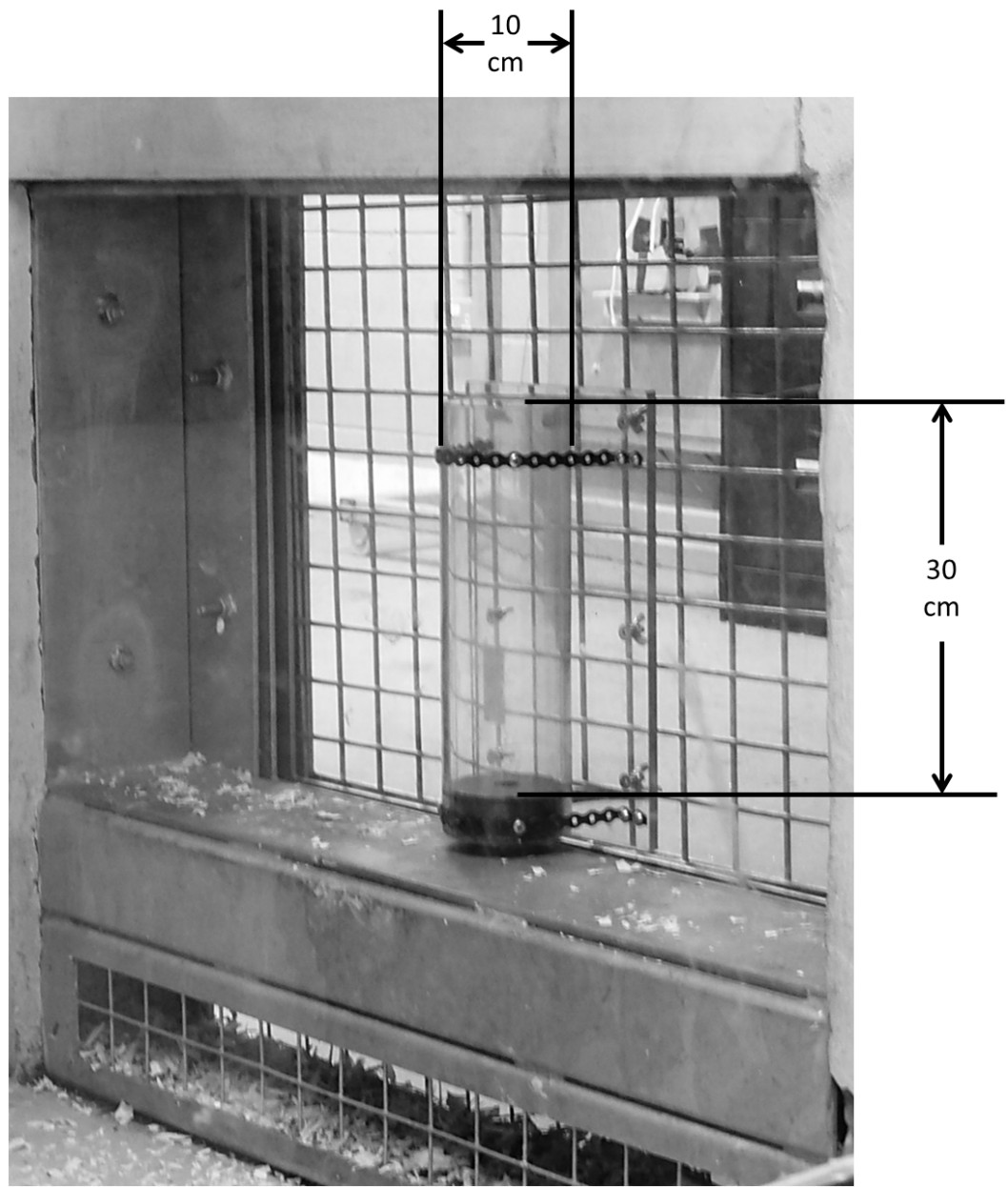

**Figure 1  Image of the apparatus as presented to chimpanzees.** The 'wide' tube as presented to the chimpanzees within the research pod. The 'narrow' tube was presented in the same manner. The height of both tubes was 30 cm, with the 'wide' tube diameter measuring 10 cm and the 'narrow' tube diameter measuring 5 cm.

filled and emptied by the experimenter through holes in the screen. Alongside the tubes, chimpanzees were provided with a selection of tool materials comprising plain sticks (rigid sticks measuring approximately 40–50 cm), straw bedding material, strips of cloth and 'wood wool'/'excelsior' (fine wood slivers typically provided as bedding material). Leafy sticks (a selection of browse generally provided to the chimpanzees by care staff for

nesting and feeding, primarily eucalyptus—these leafy sticks were flexible and measured approximately 40–80 cm) were available in the chimpanzees' enclosure throughout testing, and were placed in the research pod along with the other tool materials from the fifth hour of testing onwards. Chimpanzees were familiar with all tool materials, as straw, cloth, wood wool and browse were generally provided as nesting material, and sticks were readily available in their outdoor enclosure. All tool materials were placed into the research pods before chimpanzees were given access.

## Procedure

The apparatus was presented during twice-daily research sessions facilitated by Edinburgh Zoo staff. These sessions lasted for 45–60 min. In the first stage of the study ('Wide Tube' phase), chimpanzees were provided with the wide tube, filled with diluted sugar free Ribena to a depth of around 7 cm. This was presented alongside the aforementioned selection of tool materials, and in the 'Wide Tube' phase the juice could also be reached by hand. All provided tool materials could potentially be used to successfully retrieve juice from the task. The tube was baited with juice for ten sessions, each lasting between 45 and 60 min, with juice added by the experimenter to ensure a reward was present for the duration of each session. The tube was emptied of liquid by the researcher through a valve at the base at the end of each session, as removing the chimpanzees from the research pods was logistically unfeasible, and so the juice reward was removed in order to discourage further interaction with the tube outside of the twice-daily research session.

In the second stage of the study ('Narrow Tube' phase), chimpanzees were presented with the narrow tube, filled with juice to a depth of 7 cm, along with the same selection of tool materials. Again, chimpanzees had access to the baited tube for ten 45–60 min sessions. The narrow tube prevented chimpanzees from inserting their hands into the tube to gain juice, and also rendered attempts made using rigid, leafless sticks relatively unsuccessful, due to the ledge surrounding the panel to which the tube was attached. This ledge meant that rigid sticks inserted into the tube frequently hit the back of the tube and could be inserted no further, preventing them from reaching the juice. Leafy sticks remained functional. Absorbent materials (straw, cloth, and wood wool) could no longer be inserted and retrieved by hand.

In the third stage of testing ('Narrow Scaffolded' phase), chimpanzees were again presented with the narrow tube for ten 45–60 min sessions, but at the start of each day of testing, prior to the chimpanzees having access to the research pods, a leafy stick was inserted into the tube, with its leaves in the juice. As multiple sessions were conducted on some testing days, with no access to the research pods to re-insert the stick following the start of the first session each day, this provided four sessions which began with a stick already inserted. Only the first individual to interact with the task on these four sessions encountered this scaffolding; however, their interactions with the scaffolding and their subsequent interactions with the task could be observed by other group members.

In the course of testing, on 12 occasions, two sessions occurred on the same day, and due to the impossibility of removing chimpanzees from the research pods between sessions, chimpanzees had access to the empty tube and remaining tool materials for approximately

**Table 2  Techniques successfully used to solve the task.**

| Technique | Description | Latency to first successful use from start of first testing session | Effective in 'Wide Tube' phase? | Maintained efficacy in 'Narrow Tube' phase? | Effective in all phases? |
|---|---|---|---|---|---|
| Hand dip | Dip hand directly into juice | 00:00:35 | Yes | No | Partially effective |
| Stick dip | Stick is dipped directly into the juice | 00:00:43 | Yes | No | Partially effective |
| Stick retrieve | Stick already in tube removed with hand | 00:23:38 | Yes | No | Partially effective |
| Stick drop | Drop stick into tube then retrieve by hand | 00:21:16 | Yes | Yes | Always effective |
| Leafy stick dip | Dip stick end of a leafy branch directly into the juice (this differs from 'stick dip' only in the material) This technique remained Effective in Narrow tube phases due to the flexibility of the leafy sticks in comparison to standard sticks | 02:47:23 | Yes | Yes | Always effective |
| Leaf squash | A leafy branch is inserted stick first, and the leaves are then forced down into the tube into contact with the juice (*Note*: this technique was used successfully only once) | 05:56:26 | Yes | Yes | Always effective |
| Wood wool retrieve | Wood wool pushed part way into tube by a previous individual removed with hand (*Note*: this technique was used successfully only once) | 11:43:12 | Yes | No | Partially effective |

Notes.
The change in cell shading indicates transition between the 'Wide Tube' (light blue) and 'Narrow Tube' (dark blue) phases. The 'Narrow Tube' phase began at 8:17:44 (the start of the 11th experimental session).

one hour between sessions. It was not possible to document any attempts made during this time as the researcher did not have access to the adjacent research office to film or observe behaviours in between sessions.

## Data collection and coding

Sessions were recorded using a Sony Handycam DCR-SX21 camera. This recorded chimpanzees' responses to the task, while the experimenter narrated chimpanzees' actions as well as the identity and presence of other individuals in the research pod. The video and narration were later coded together. The chimpanzee identity, tool material choice (hand, plain stick, wood wool, straw, or leafy stick), action (dip, retrieve, squash, drop), and success of individuals attempting the task were coded for each attempt. The presence of other individuals in the research pod was also coded for each attempt. Techniques were then classified as 'Always effective' and 'Partially effective' (Table 2), based upon their potential efficacy in narrow tube conditions. A second coder, blind to the experimental hypotheses, coded 50 randomly selected attempts for tool material, technique (as listed in Table 2), and success. Narration of these attempts did not include narration of tool material, technique or success, and so the second coding was independent. Inter-observer reliability was calculated using Cohen's Kappa, revealing agreement for all variables (tool material $K = 0.90$, technique $K = 0.87$, success $K = 0.95$).

## Data analysis

Data were analysed in R (version 3.2.2, *R Core Team, 2015*) and RStudio (version 0.99.893, *RStudio Team, 2015*) using a generalised linear mixed model with a Laplace approximation (GLMM), using the function glmer in the R statistics package lme4 (*Bates, Maechler & Bolker, 2012*). Analysing binomial data using GLMM is recommended by *Jaeger (2008)* and *Bolker et al. (2009)*. The impact of predictor variables upon the number of 'Always effective' vs 'Partially effective' technique attempts in the 'Wide Tube' and 'Narrow Tube' phases was assessed for all individuals that made attempts in both the 'Wide' and 'Narrow' tube phases ($N = 10$).

A full random slope model was fitted (for discussion on the benefits of fitting random slopes as well as random intercepts, see *Schielzeth & Forstmeier, 2009*). The full model contained fixed effects for Phase ('Wide Tube' vs 'Narrow Tube') and Subgroup (Edinburgh vs Beekse Bergen), along with an interaction between the two and a random intercept and slope by Phase for each Individual ($N = 10$). This full model was compared with a null model including only the random intercept and slope by Phase for each Individual. Both models were fitted using a binomial error structure, due to the binary nature of the response variable (Effectiveness), and a logit link function. A likelihood ratio test comparing the full and null models indicated that the full model was a significantly better fit ($\chi^2 = 8.65$, $df = 3$, $p = .0343$; dAIC = 2.7).

In addition, a full random slope GLMM was fitted to examine the likelihood of 'Always effective' vs 'Partially effective' across experimental sessions within the 'Narrow Tube' phase, in order to investigate any change in behaviour over time within this phase. The model contained a fixed effect for Session (the ten 45–60 min sessions that comprised each phase), and a random intercept and slope by Session for each Individual that took part in the 'Narrow Tube' phase ($N = 11$). As with the main analysis described above, this model was compared with a null model containing only the random effects, and a likelihood ratio test indicated that the model containing a fixed effect for Session was no improvement upon a null model ($\chi^2 = 2.37$, $df = 1$, $p = .1238$; dAIC = 0.4).

## RESULTS

Sixteen of the eighteen chimpanzees interacted with the task during at least one of the three phases of testing (see Table 1 for frequency of participation for all individuals) In addition, sixteen of eighteen individuals were present in the research pod during another individual's attempt at the task on at least one occasion (with only Cindy and Liberius never having the opportunity to observe others at the task). Ten individuals attempted the task in both the 'Wide Tube' and 'Narrow Tube' phases. Seven individuals participated in all three phases (see Table 1). A total of 3,022 attempts were made across the three phases and 30 h of testing. Seven techniques were used to successfully solve the task (see Table 2). Six of these techniques emerged in the 'Wide Tube' phase (and four of these six within the first testing session), with the remaining technique emerging in the 'Narrow Tube' phase. No novel techniques were observed in the 'Narrow Scaffolded' phase. Techniques involving the use of hands only and the use of plain, rigid sticks emerged first, within the first testing session.

**Table 3** **The two most frequently used techniques of individuals that made attempts in both the 'Wide' and 'Narrow' tube phases.** 'Always effective' techniques are shown in bold. An asterisk indicates that an individual used a technique for the first time within the 'Narrow' Tube phase (i.e., that the technique was novel to them).

| Subgroup | Individual | 'Wide Tube' phase | | 'Narrow Tube' phase | |
|---|---|---|---|---|---|
| | | 1st preferred technique (no. of attempts; % of attempts) | 2nd preferred technique (no. of attempts; % of attempts) | 1st preferred technique (no. of attempts; % of attempts) | 2nd preferred technique (no. of attempts; % of attempts) |
| Edinburgh | Kindia | Stick dip (6; 43%) | **Leafy stick dip** (6; 43%) | Stick dip (2; 50%) | **Leafy stick dip** (2; 50%) |
| | Louis | Stick dip (6; 100%) | | Stick dip (13; 87%) | ***Leafy stick dip** (2; 13%) |
| | Lucy | Stick dip (7; 64%) | Hand dip (4; 36%) | Stick dip (2; 100%) | |
| | Kilimi | Stick dip (127; 73%) | Hand dip (33; 19%) | Stick dip (49; 48%) | **Leafy stick dip** (44; 43%) |
| | Emma | Stick dip (19; 51%) | Hand dip (16; 43%) | Stick dip (38; 64%) | Hand dip (8; 14%) / ***Leafy stick dip** (8; 14%) |
| Beekse Bergen | Pearl | Hand dip (165; 53%) | Stick dip (134; 43%) | Stick dip (119; 56%) | **Leafy stick dip** (71; 33%) |
| | Edith | Stick dip (320; 83%) | Hand dip (64; 17%) | Stick dip (364; 49%) | ***Leafy stick dip** (294; 40%) |
| | Eva | Stick dip (237; 76%) | Hand dip (61; 20%) | Stick dip (12; 57%) | Hand dip (5; 24%) |
| | Frek | Hand dip (74; 59%) | Stick dip (27; 22%) | Stick dip (40; 50%) | **Stick drop** (23; 29%) |
| | Rene | Hand dip (141; 93%) | Stick dip (7; 5%) | Hand dip (6; 55% ) | Stick dip (3; 27%) |

**Notes.**
Percentages are rounded.

The use of leafy sticks emerged after several hours of exposure to the task (Table 2). Only one novel technique (*wood wool retrieve)* emerged in the 'Narrow Tube' phase, though some individuals performed techniques in this phase that were novel to them (though not novel to the group, and so potentially acquired socially) (Table 3).

## Behavioural flexibility

In order to examine behavioural flexibility in the chimpanzees' response to the task restrictions imposed by the transition from the 'Wide Tube' to 'Narrow Tube' phase, the techniques described above were categorised as 'Always effective' or 'Partially effective' according to their potential efficacy across phases (see Table 2). Note that all techniques (see Table 2) were effective in the 'Wide Tube' phase, as the tube width allowed individuals to insert their hands into the tube, facilitating the use of a wide variety of techniques.

Techniques involving the insertion of a hand into the tube (i.e., *hand dip*) were classified as 'Partially effective', as insertion of the hand into the tube was made impossible by the width of the tube in the 'Narrow Tube' phase, though this technique could be successfully used in the 'Wide Tube' phases. Techniques reliant upon the insertion of a plain, rigid stick into the tube were also classified as 'Partially effective', as the overhang of the task presentation window was such that these sticks could frequently no longer be inserted into the tube in the 'Narrow Tube' phase. Short rigid sticks could still successfully be dropped into the tube and retrieved by hand, and so the technique *stick drop* is categorised as 'Always effective'. For further discussion of this classification, see Article S1.

Comparison of the use of 'Partially effective' and 'Always effective' techniques in the 'Wide Tube' and 'Narrow Tube' phases indicates the extent to which chimpanzees altered their behaviour in response to the change in task, and the extent to which they were able

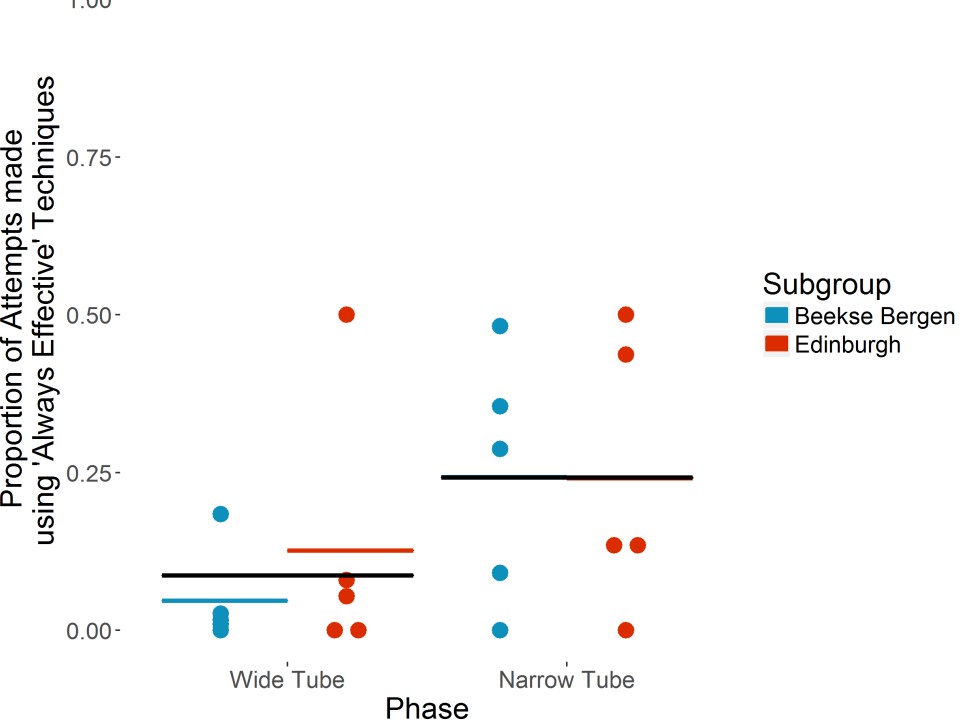

**Figure 2** **Proportion of attempts made using 'Always effective' techniques in 'Wide' vs 'Narrow' tube phases by the ten chimpanzees that took part in both phases.** Individual chimpanzees' proportions are indicated by dots. Coloured horizontal bars show each subgroup's mean proportion. Black horizontal bars show overall mean proportion. Note that in the 'Narrow Tube' phase, the subgroup means and overall mean are the same.

to set aside techniques that had been rendered impossible by the change in tube width. An increase in the use of 'Always effective' techniques in the 'Narrow Tube' phase compared with the 'Wide Tube' phase would indicate a flexible response to the task changes. The 'Narrow Scaffolded' phase is not included in these analyses, in order to exclude the possibility of the scaffolding impacting observed flexibility.

The ten individuals that took part in both 'Wide' and 'Narrow' tube phases used 'Always effective' techniques for 58 of 1,527 attempts in the 'Wide Tube' phase, increasing to 514 of 1,250 attempts in the 'Narrow Tube' phase. On average, each individual used 'Always effective' techniques for a mean of 8.7% (SD = 15.6) attempts in the 'Wide Tube' phase, increasing to a mean of 24.2%% (SD = 19.4) of attempts in the 'Narrow Tube' phase (see Fig. 2). Individual preferences in terms of specific techniques for the 10 individuals that took part in both 'Wide' and 'Narrow' tube phases are shown in Table 3.

The full model (see Table 4) indicates that use of 'Always effective' tool techniques increased significantly in the 'Narrow Tube' phase ($b = 2.93$, $p = .0013$); in the 'Narrow Tube' phase, chimpanzees were 18.67 (Wald 95% CI [3.16–110.47]) times more likely to use an 'Always effective' technique than in the 'Wide Tube' phase. This indicates a significant increase in 'Always effective' technique use in the 'Narrow tube' phase, and as

**Table 4  Results of full model GLMM on the effects of Phase, Subgroup and an interaction between the two, with random intercept and slope for Individual by Phase upon 'Always effective' technique use.** Variance, standard deviation, and correlation for the random intercept and slope for Individual by Phase is provided.

| Fixed effects | Estimate | [Wald 95% CI] | Std. Error | z value | p value |
|---|---|---|---|---|---|
| Intercept (including 'Wide Tube' phase and Beekse Bergen subgroup) | −4.13 | [−5.62, −2.64] | 0.76 | | |
| Phase ('Narrow Tube') | 2.93 | [1.15, 4.70] | 0.91 | 3.23 | 0.0013** |
| Subgroup (Edinburgh) | 1.57 | [−0.63, 3.77] | 1.12 | 1.40 | 0.1631 |
| Phase*Subgroup | −1.57 | [−4.17, 1.04] | 1.33 | −1.18 | 0.2387 |

| Random effects | Variance | Std. Deviation | Correlation |
|---|---|---|---|
| Individual (Intercept) | 2.46 | 1.57 | |
| Slope by Phase | 3.31 | 1.82 | −0.87 |

**Notes.**
** $p < 0.01$.

we present a binomial GLMM, and each attempt made could only be 'Always effective' or 'Partially effective', an equivalent significant decrease in the use of 'Partially effective' techniques.

There was no significant effect of Subgroup ($b = 1.57$, $p = .1631$) upon the likelihood of an individual using 'Always effective' techniques in either Phase, and no significant interaction between Phase and Subgroup ($b = -1.57$, $p = .2387$), indicating that the two Subgroups did not respond significantly differently to the change in Phase. A likelihood ratio test indicated no significant difference between this full model (including the impact of Subgroup and an interaction with Phase) and a reduced model including only Phase ($\chi^2 = 1.52$, $df = 2$, $p = .4676$; dAIC = 2.4), and so we present the full model with all predictors here.

An additional analysis examining the impact of Session upon the likelihood of individuals using 'Always effective' techniques within the 'Narrow Tube' phase found that Session had no significant effect ($b = 0.17$, $p = 0.077$), indicating that individuals did not become more likely to use 'Always effective' techniques as time passed within the 'Narrow Tube' phase.

### "Narrow Scaffolded" phase

Four individuals (Kindia, Edith, Frek, and Pearl) encountered the task with the scaffolded leafy stick solution in place. Of these individuals, only two (Edith and Pearl) put the leaves in their mouths to retrieve the reward. Frek and Kindia instead discarded the branch without retrieving any reward from it. These four individuals (Kindia, Edith, Frek, and Pearl) who encountered the leafy stick solution first-hand did not use the leafy stick solution (dipping the top, leafy part of the branch in, rather than the bare, stick end). It therefore appears that scaffolding the behaviour in this limited manner did not provide sufficient information for these chimpanzees to acquire a novel 'Always effective' solution to the task.

### DISCUSSION

In this study, chimpanzees were found to be capable of responding with some flexibility to a changing task. However, although individuals significantly increased their use of

'Always effective' techniques in the 'Narrow Tube' phase, no individual that made more than one attempt used 'Always effective' techniques for a majority of their attempts. This indicates that chimpanzees' behavioural flexibility was limited by an inability to relinquish the previously successful solutions used in the 'Wide Tube' phase, which continued to comprise at least 50% of all individuals' attempts. Scaffolding provided to four individuals in an attempt to facilitate use of an 'Always effective' technique in the final 'Narrow Scaffolded' phase did not result in the use of this novel technique by this limited subsample of chimpanzees. Membership of subgroups with differential experiential histories did not have a significant impact upon use of 'Always effective' techniques. The principal issues addressed by the study are discussed in turn below.

## Behavioural flexibility

Chimpanzees did typically alter their behaviour in response to the change in task, using significantly more 'Always effective' techniques in the 'Narrow Tube' phase. However, no individual that made more than one attempt used 'Always effective' techniques for a majority of their attempts, in contrast with the findings of *Manrique, Völter & Call (2013)* in whose study chimpanzees successfully used an effective solution for the majority of attempts in the appropriate condition. Using a very similar task to the current study, *Lehner, Burkart & Van Schaik (2011)* found that orangutans preferentially used efficient methods when task constraints rendered their previous preferences inefficient. Similarly, *Davis et al. (2016)* found that most chimpanzees used a novel, efficient solution for the majority of attempts when the efficiency of their previously known solution became very low, though this extent of switching was observed only in groups with a trained demonstrator using the efficient solution or with human demonstrations provided: in 'non-seeded' groups, only one individual switched to the efficient solution, but then used it for the majority of attempts. These comparisons with previous research indicate that whilst chimpanzees in this study showed behavioural flexibility (altered their behaviour), the flexibility was limited in comparison to several previous studies, thus concurring with some other previous studies of chimpanzees (*Marshall-Pescini & Whiten, 2008*; *Hrubesch, Preuschoft & Schaik, 2009*; *Bonnie et al., 2012*). The finding also concurs to some extent with experimental studies of wild chimpanzees which have shown that individuals approach novel problems with culturally-informed known behaviours, which may limit the extent to which individuals perceive the affordances of alternative tools (*Gruber et al., 2009*; *Gruber et al., 2011*). However, evidence from wild observations also shows that chimpanzees are capable of modifying tool behaviour (*Yamamoto et al., 2008*), and using novel materials to achieve known forms of tool use (*Hobaiter et al., 2014*), both of which indicate some level of behavioural flexibility in individual wild chimpanzees.

Regarding the 'explore-if-dissatisfied' strategy discussed previously, we found that only one novel technique emerged following the imposition of task constraints (see Table 2), and this technique was used successfully only once. This would seem to indicate that chimpanzees in the current study did not respond to the task changes by exploring the possibility of novel solutions, but rather altered the extent to which they employed known solutions. Three individuals (Louis, Emma and Edith) used a technique that
was novel to them in the 'Narrow Tube' phase for a considerable proportion of their attempts (see Table 3). The technique in question (*leafy stick dip*) was not novel on a group level, and so this may be the result of observational learning (perhaps indicating a 'copy-if-dissatisfied' strategy). The relatively limited flexibility found in the current study could be due to differences in task demands between the present study and others reporting more flexibility. One possible explanation for the difference in the apparent level of flexibility seen in this study compared with *Davis et al. (2016)* is the presence in the *Davis et al. (2016)* study of social information provided by a trained, competent model. The scaffolding in the final 'Narrow Scaffolded' phase aside, chimpanzees were not provided with trained conspecific demonstrators or human demonstrations to offer social information about 'Always effective' techniques. Had the chimpanzees in this study entirely failed to discover 'Always effective' techniques, this lack of experimentally-provided social information would be a plausible explanation for the relatively diminished behavioural flexibility observed. However, the majority of individuals tested used 'Always effective' techniques in the 'Narrow Tube' phase, and indeed had access to social information in the form of others using 'Always effective' techniques, indicating that they were not limited by a lack of knowledge of the existence of such techniques, but rather failed to employ them as frequently as they used 'Partially effective' techniques.

As in the current study, apes in *Manrique, Völter & Call (2013)* and *Lehner, Burkart & Van Schaik (2011)* had to discover more efficient solutions to the provided tasks without additional information from experimenters (though in both *Lehner, Burkart & Van Schaik, 2011*, and the current study, apes were tested in a group context and thus had access to any social information conspecifics provided). Though three individuals did use an 'Always effective' technique that was novel to them in the 'Narrow Tube' phase (Table 3), indicating a capacity for acquisition of novel behaviours via either social or individual learning in response to the change in task, only three of the seven observed solutions (*leafy stick dip*, *leaf squash*, and *stick drop*) were 'Always effective' in the 'Narrow Tube' phase. This limited range of available solutions may have restricted chimpanzees' ability to respond flexibly to the change in task. In comparison, orangutans in the *Lehner, Burkart & Van Schaik (2011)* study exhibited six effective solutions in 'Restricted Condition 1' (analogous to our 'Narrow Tube' phase) and so had knowledge of a wider range of potential solutions that could be usefully applied to the narrow tube. Comparison of the range of solutions discovered by chimpanzees in the current study with the range of solutions observed by *Tonooka, Tomonaga & Matsuzawa (1997)* in response to a task similar to the 'Wide Tube' phase of our study may indicate that chimpanzees in our study were somewhat limited in their exploration of the task, with *Tonooka, Tomonaga & Matsuzawa (1997)* observing 16 different solutions in comparison to only seven solutions seen in the current study. Although chimpanzees in *Tonooka, Tomonaga & Matsuzawa (1997)* had access to a wider range of tool materials, which, rather than a lack of exploration, may account for the discrepancy in the number of solutions observed in their study and the current study, there were tool materials available to chimpanzees in our study (namely *cloth* and *wood wool*) that were never used in the 'Wide Tube' phase, when there were few limitations to the efficacy of solutions.

The fact that chimpanzees in our study did not explore the potential of absorbent materials as tool solutions to the task could suggest a role for 'functional fixedness' (*Hanus et al., 2011*) in limiting their exploration of the task. While the chimpanzees are familiar with these absorbent materials, they are primarily used as nesting material. The chimpanzees may therefore have struggled to perceive them as having an alternative use. This does not appear to have constrained the chimpanzees' use of leafy sticks as tools (despite such browse often also being used as nesting material), though their reluctance to use the leaves of these sticks to dip with may also reflect a level of 'functional fixedness', as only the stick portion of such leafy sticks appeared to be considered functional by the chimpanzees.

In contrast to the current study, the tasks provided by *Manrique, Völter & Call (2013)* and *Davis et al. (2016)* did not require the use of tools, and were solvable by hand. Tasks requiring tool use are typically more challenging than those requiring purely manual actions, with the inclusion of tool use more challenging for causal cognition (*Seed et al., 2009*; *Völter & Call, 2014*). The necessity of tool use in the latter conditions of this study ('Narrow Tube' phases) may therefore have impeded behavioural flexibility, perhaps by confounding the chimpanzees' comprehension of the efficacy of their solutions, or impeding the acquisition of further 'Always effective' solutions via social or asocial means.

## The impact of scaffolding

The limited exposure to scaffolding provided to four chimpanzees in the third and final phase of this study did not lead to the acquisition of novel techniques by any individual. This scaffolding (providing the task with a leafy stick already inserted) aimed to approximate a form of information chimpanzees have access to in the wild—the products and debris of other chimpanzees' tool use, hypothesised to be a facilitator of learning local techniques by *Tennie, Call & Tomasello (2009)* (e.g., "…nut crackers and termite fishers leave their tools and detritus behind, and in the right place, which makes the learning of their offspring and others much easier", pp.2406). This information in the form of residual, enduring artefacts has been argued to facilitate technical activities in New Caledonian crows (*Holzhaider, Hunt & Gray, 2010*), Japanese macaques (*Leca, Gunst & Huffman, 2010*) and bearded capuchins and chimpanzees (*Fragaszy et al., 2013*), although we are not aware of any direct experimental evidence for such hypothesised effects.

Although we are cautious in our interpretation of our results regarding scaffolding, given that only four individuals had very limited interaction with the scaffolding, more extensive efforts to provide this kind of information in experimental settings have often proven unsuccessful in encouraging the invention of novel behaviours. Wild bearded capuchin monkeys from a population that does not habitually use probing tools, when presented with a dipping task with stick probes already inserted, never acquired probing behaviour (*Cardoso & Ottoni, 2016*). Using a similar task to the current study, *Gruber et al. (2009)* found that two communities of wild chimpanzees had distinct tool use preferences when extracting honey from a cavity, with one community solving the task using a stick tool while the other solved it using their hands or leaf sponges. Subsequently providing this task to the hand/leaf-sponge community with a stick tool already inserted failed to elicit stick tool use (*Gruber et al., 2011*). The authors interpret this finding as evidence of

"cultural bias" towards existing traditions of tool use, which constrains how individuals perceive and evaluate the affordances of their environment (*Gruber et al., 2011*), but it may also be that presenting chimpanzees with scaffolded solutions in this manner (at least when numbers of such presentations are limited, with the majority of chimpanzees in Gruber et al.'s study receiving only one or two exposures) simply provides insufficient information to elicit acquisition of novel tool use behaviours. *Gruber et al. (2011)* argue that such limited exposure may be ecologically valid, and tests the hypothesis that novel tool behaviours will emerge in direct response to favourable affordances (a situation that highlights the physical properties of potential tool materials and indicates the potential relationship between tool and goal), but the possibility remains that such limited exposure to scaffolding is insufficient to impact behaviour.

Similarly, experiments incorporating 'ghost conditions', in which the operational affordances of a task are demonstrated without the presence of a demonstrator individual (e.g., task components are moved by fine fishing line or similar), perhaps also corroborates the insufficiency of such scaffolding. *Hopper et al. (2007)* and *Hopper et al. (2015)* found that demonstration of the workings of a 'Panpipes' apparatus in such a ghost condition did not lead chimpanzees to discover the solution to the task. The efficacy of such ghost demonstrations may be affected by the complexity of the task (*Hopper et al., 2008*), with complex tool use perhaps proving more challenging to learn via such demonstrations. The information provided by these scaffolded conditions (as the current study, and *Gruber et al., 2011*) can also be argued to be even more impoverished than that provided by ghost conditions as it does not incorporate movement, and so may not demonstrate the affordances of a task to the same extent as moving displays.

Providing further groups of chimpanzees with scaffolding approximating the artefacts of other's tool use they might encounter in the wild, perhaps in the context of novel tool use tasks for which no pre-existing cultural biases are likely to exist, could shed further light on the ability of chimpanzees to learn novel behaviours from this sort of information, and indicate whether chimpanzees are limited by cultural biases, by the relative poverty of the information provided by scaffolding, or simply require more extensive exposure to scaffolding in order to acquire novel behaviours.

## Subgroup differences

Subgroup membership was found to have no significant effect on the likelihood of individuals using 'Always effective' techniques in either the 'Wide' or 'Narrow' tube phase, with both groups increasing their use of 'Always effective' techniques in the 'Narrow' tube phases. Previous research has found that chimpanzees reared in captivity perform more poorly than wild-born chimpanzees in tool-use tasks. *Morimura & Mori (2010)* found that captive-reared chimpanzees were less likely to succeed than wild-born chimpanzees on a tool use task involving retrieving juice from a bottle using a provided stick tool. Similarly, *Brent, Bloomsmith & Fisher (1995)* found that both captive mother-reared and captive nursery-reared chimpanzees were less likely to succeed than wild-born chimpanzees (in captivity) on a reaching tool task. Rearing history has also been found to have an impact on nest-building, with wild-born chimpanzees in captivity spending more time constructing

and using nests than captive-born chimpanzees, and using more complex techniques in their nest building, while within the captive-born chimpanzee sample, mother-reared individuals spent more time than nursery-born individuals making and using nests (*Videan, 2006*). Some research indicates that housing can influence performance on cognitive tests, with *Vlamings, Hare & Call (2010)* finding that sanctuary-housed chimpanzees and bonobos outperform zoo-housed apes in a detour-reaching task testing inhibitory control. However, *Wobber & Hare (2011)*, using a subset of the Primate Cognition Test Battery (*Herrmann et al., 2007*), found no impact of housing (zoo vs sanctuary) on the performance of mother-reared chimpanzees on tests of social or physical cognition, but did find that mother-reared individuals (regardless of housing) outperformed orphaned individuals on physical cognition tests—this effect was due to improved performance on a tool properties task in which individuals chose between a functional and non-functional tool. Similarly, *Herrmann, Wobber & Call (2008)* found no difference in performance between zoo- and sanctuary-housed chimpanzees and orangutans on a tool functionality choice task.

We did not find evidence of subgroup differences in the current study, though the subgroups compared here were based upon the chimpanzees' housing history rather than their rearing. Although only the Beekse Bergen group includes individuals hand-reared by humans, and this is the most common rearing history for this subgroup (five of nine individuals), both groups include both wild- and captive-born mother-reared individuals. Sample size precluded any analysis based upon rearing history (with only three individuals categorised as wild-born, only two of which participated in both the 'Wide' and 'Narrow Tube' phases of the task), and given the difference in origin of the two groups (laboratory vs. zoo) it was thought a subgroup analysis might capture any differences in performance related to early life experiences. It may be that, between captive environments housing chimpanzees with mixed rearing histories, there is little difference in performance to be found using artificial foraging tasks such as the current task. However, further groups of chimpanzees in different captive facilities would have to be provided with the task in order to test this. Examination of the impact of both housing and rearing histories upon chimpanzee performance in experimental tasks is essential for a full understanding of chimpanzee cognition (*Boesch, 2007*).

## Limitations to the current study

Although some previous studies of chimpanzee and great ape behavioural flexibility have tested individuals in isolation (*Marshall-Pescini & Whiten, 2008*; *Manrique, Völter & Call, 2013*; *Manrique & Call, 2015*), and this approach allows for control over chimpanzees' access to social information about the task and the avoidance of issues around task monopolization by dominant individuals, we note that many previous studies have presented tasks in a group context (*Gruber et al., 2009*; *Gruber et al., 2011*; *Lehner, Burkart & Van Schaik, 2011*; *Davis et al., 2016*). This may be due to the constraints of conducting research at captive facilities (or, in the case of *Gruber et al., 2009*; *Gruber et al., 2011*, in the wild), or be a methodological choice in order to explore the social learning abilities of subjects (*Davis et al., 2016*). The open presentation of the task in the current study to the group of chimpanzees prevents us from making claims regarding the process by

which novel behaviours emerged and spread in individuals during the trials (whether this occurred via social learning or innovation). However, following *Lehner, Burkart & Van Schaik*'s (*2011*) definition of behavioural flexibility, as the continued acquisition of novel techniques via either social or asocial learning, group testing in this manner allows flexibility to be demonstrated through either form of learning.

This study focused upon a community of chimpanzees with particular life histories (primarily captive born) and was also limited to adult chimpanzees, with no infants or juveniles in the population. Previous research has shown that behavioural flexibility (referred to as 'cognitive flexibility' by *Manrique & Call, 2015*, and measured by a simple reversal task) follows a U-shaped curve in great apes, with optimum performance in terms of error avoidance appearing to occur between seven and 27 years of age (*Manrique & Call, 2015*), and so the age of some individuals in the current study may have impaired their performance (with five of the ten individuals that provided data in both the 'Wide' and 'Narrow Tube' phases being over 27 years old). Position in the dominance hierarchy could also impact performance on an artificial foraging task such as that of the current study. Low ranking individuals may be more likely to innovate novel behaviours (*Reader & Laland, 2001*), and in a group context, dominance may impact the likelihood of others acquiring a behaviour from an individual via social learning (*Horner et al., 2010*; *Kendal et al., 2015*, though see also *Watson et al., 2017*). With a sample of only ten individuals providing data across the 'Wide' and 'Narrow Tube' phases, analyses of age, rearing history, and rank effects were not possible in the current study, but these previous findings demonstrate the importance of considering the impact of such factors upon task performance when possible. Future work would benefit from expanding the sample to include a more diverse range of ages and rearing histories, in sufficient numbers to enable analysis of the potential effects of these and other factors upon participation and performance in artificial foraging tasks.

## Conclusions

Chimpanzees in this study did respond flexibly to a changing task, increasing their use of 'Always effective' techniques when task alterations rendered previously used solutions unrewarding. However, no individual altered their behaviour to the extent of using 'Always effective' solutions for a majority of attempts, indicating relatively limited behavioural flexibility in comparison with some previous research (*Lehner, Burkart & Van Schaik, 2011*; *Manrique, Völter & Call, 2013*; *Davis et al., 2016*) and a tendency to continue to use previously rewarded behaviours. The relatively limited flexibility observed here may be due to the complexity of the task, which required tool use. A subset of chimpanzees did not acquire a more effective, novel, tool technique when provided with a single instance of scaffolding towards the solution.

Research on behavioural flexibility in chimpanzees, both in the wild and in captivity, using a variety of artificial foraging tasks, continues to provide divergent results, with some studies indicating strong conservatism, while others find an ability to relinquish known solutions in favour of more rewarding or efficient techniques. Further work must been done to investigate what factors limit or encourage flexibility in great apes, with

promising avenues for further study being the relative complexity of the task (for example, the requirement for tool use versus tasks which can be solved by hand), the disparity in efficacy between known and alternative techniques (whether the established solutions become entirely unrewarded or simply less efficient or rewarding than an alternative) and investigation of individual characteristics such as age or rank which may promote or constrain behavioural flexibility.

## ACKNOWLEDGEMENTS

We are grateful to the Royal Zoological Society of Scotland for permission to conduct this study at Edinburgh Zoo, and to the Budongo Trail keepers for all their assistance. We thank Tom Houslay for statistical discussion and advice, Emily Kilkenny for coding, and Kim Bard for comments on an earlier version of the manuscript.

### Funding

This work was supported by a John Templeton Foundation grant ID 40128 to Andrew Whiten. The funders had no role in study design, data collection and analysis, decision to publish, or preparation of the manuscript.

### Grant Disclosures

The following grant information was disclosed by the authors:
John Templeton Foundation: 40128.

### Competing Interests

The authors declare there are no competing interests.

### Author Contributions

- Rachel A. Harrison conceived and designed the experiments, performed the experiments, analyzed the data, prepared figures and/or tables, authored or reviewed drafts of the paper.
- Andrew Whiten conceived and designed the experiments, authored or reviewed drafts of the paper.

### Animal Ethics

The following information was supplied relating to ethical approvals (i.e., approving body and any reference numbers):

The study received ethical approval from the University of St Andrews Animal Welfare and Ethics Committee, and was approved by the Budongo Trail Research Committee. Research was conducted in accordance with the guidelines of the Association for the Study of Animal Behaviour.

### Data Availability

Raw data is available in the Supplemental Information.

## Supplemental Information

Supplemental information for this article can be found online at http://dx.doi.org/10.7717/peerj.4366#supplemental-information.

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
