# Peer review of "Chimpanzees (Pan troglodytes) display limited behavioural flexibility when faced with a changing foraging task requiring tool use"

_PeerJ, doi:10.7717/peerj.4366_

## Round 0.1 · original submission · Major Revisions

I was extremely fortunate to obtain reviews from three experts who were both timely and thorough with their reviews. They have all converged upon the opinion that your study is of great interest, but that its presentation could benefit from a clearer definition of the key concepts and a more detailed background and rationale. I too am very interested in the topic of your paper but feel that it could be more focused. The reviewers have all recommended major revisions and make several very useful suggestions for improving the paper’s clarity and impact. I am happy to invite you to submit a revision that incorporates these suggestions. I agree that you should be clearer about how you expect the subgroups to differ based on their differing histories. Could you provide more information about the prior experimental history of both groups? Also please provide reliability analyses for the coded data.

I have a few minor suggestions of my own:

Please change “which” to “that” on lines 28 and 32 of the abstract. See line 38 as well and elsewhere.
I am curious whether the definition of behavioral flexibility must include the phrase “based upon environmental feedback” (line 37). I can imagine a situation in which someone is intrinsically motivated to attempt different behaviors (perhaps out of boredom) without any prompt from the environment per se.
Around lines 85-88, you might mention that chimpanzees also sometimes persist in actions that are no longer necessary (e.g., the trab tube problem where they continued to follow a rule about where to insert the stick even when the trap was ineffectual, Povinelli, 2000). Of course humans sometimes do this as well (Silva & Silva, 2005, 2006).
On line 115, please move the “only” to after “apparent”.
On line 121, the phrase “Given this research background” is unnecessary and can be deleted.
Could you examine sex differences or effects of dominance status in your sample? I think the treatment of the data needs to take into account the influence of other individuals’ actions because of the group testing situation and I agree with the reviewers that more needs to be said about potential advantages and disadvantages of group testing.
Thank you for submitting such an interesting and well-written paper to PeerJ.

Reviewer 1 ·

Basic reporting

No comment.

Experimental design

I have one major concern (pertaining to the authors’ objectives and hypothesis-testing framework… or lack thereof) that needs to be addressed before I can recommend this manuscript for publication.
A couple of minor specific comments should also be addressed.

Validity of the findings

No comment.

Comments for the author

This study aimed to investigate the capability of chimpanzees to alter their behaviour in response to an artificial foraging task (i.e., liquid-retrieval task) in which viable solutions became gradually restricted in captive chimpanzees originating from two separate subgroups with different experiential histories, and after providing them with limited exposure to scaffolding towards a novel tool technique.

Overall, this is a well-written manuscript on an interesting topic with key implications for our understanding of the evolution of cumulative culture in our hominid ancestors. The Methods, Results, and Discussion sections are strong and well laid out.

However, I have one major concern (pertaining to the authors’ objectives and hypothesis-testing framework… or lack thereof) that needs to be addressed before I can recommend this manuscript for publication.
A couple of minor specific comments should also be addressed.

MAJOR CONCERN:

I think the “study aims” section (Line 120-134) should take a more robust hypothesis-testing approach. In the present version of the manuscript, this section is much too general, descriptive and vague (“This may be anticipated to impact their problem-solving abilities”; “this study also considers the impact of subgroup membership upon task performance”; “we examined whether chimpanzees could acquire novel tool behaviour through exposure to favourable affordances”).
After providing a very interesting background review of the literature on behavioral flexibility and conservatism in tool-use and tool-making tasks, I am wondering why the authors did not use some of the theoretical aspects they introduced (e.g., the “copy-if-dissatisfied” social learning strategy) and the potential asset of their study conditions (i.e., a group of subjects originating from two different subgroups with different experiential histories) to generate at least two competing/alternative hypotheses (e.g., behavioral flexibility versus behavioral conservatism… or functional fixedness), each of them being associated with a series of testable and mutually exclusive predictions. Without this framework, I started reading the Methods, Results, and Discussion sections with no specific expectations in mind, which, I believe, may seriously weaken the impact of the study.

MINOR CONCERNS:

To what extent the authors’ results could be interpreted in terms of “functional fixedness”, which is not mentioned in the paper?
As a reminder, functional fixedness is the disinclination to use familiar objects (including tools) in novel ways, and as such, is a likely inhibitory factor in tool innovation? (cf. Brosnan and Hopper, 2014).

Line 381: “although we are not aware of any direct experimental evidence for such hypothesised effects.” – Please read the fascinating observational and experimental studies of the developmental processes and indirect social influences (via behavioral artefacts) underlying the acquisition tool use and tool-making techniques (Pandanus tools) in wild New Caledonian crows, and showing that juvenile birds learn these complex behaviors by manipulating parents’ (and others’) discarded tools and using counterpart artefacts (i.e., negative template of the tool after removal from the thick and enduring Pandanus leaf) as easier starting points for tool-making (e.g., work by Jennifer Holzhaider, Gavin Hunt and Russell Gray).

·

Basic reporting

Dear authors,
This is an interesting study on the ability of individual chimpanzees to change their behavior when a task changes.

Experimental design

There are two main points I’d like more information on throughout:

1) Examine within-subject changes in behavior of effective vs not effective solutions depending on which phase the response was recorded in (see more details below).

2) Say something about why you tested them socially in the introduction and how you predict this to influence your measurement of flexibility and innovation.

Validity of the findings

My more detailed comments follow below.

Comments for the author

L33: define “conservatism” or just say what it is rather than using jargon

L39: explain a bit how flexibility supports problem solving

L41: define “culture” and “cumulative culture”

L42: say something about the tension in cumulative culture between modification of the trait through flexibility and needing to have high copying fidelity. The latter is often thought to be the main facilitator of cumulative culture (e.g., Hunt & Gray 2003).

L46: how do you predict these traits (flexibility, innovation, social learning) limit cumulative culture in chimpanzees?

L52: Logan 2015 and Logan 2016 are the same paper. The 2015 version is the preprint at bioRxiv and the 2016 version is the peer reviewed published version. You can delete the Logan 2015 citation.

L59: did the differences in flexibility support or refute predictions?

L60: why are “individual differences” important to cumulative culture and flexibility? A bit more explanation will help place these results in context so the reader can follow your argument.

L64: were individual differences not found in corvids and great apes because they weren’t investigated? Why the switch to interspecific for these taxa and what does this level of explanation contribute to furthering your argument about flexibility and cumulative culture?

L71: define conservatism

L73: point out that it might reflect the flexibility of the innovating individual(s) (but not those that subsequently used the innovators’ technique/material choice).

L82: I’m not sure how you consider innovation distinct from flexibility. It would be good to define innovation in the first paragraph of the introduction.

L101: indicate whether the great apes were successful on the multi-access box task.

L102: explain how the juice retrieval task works.

L113 and 117: in the discussion, you don’t mention what your results indicate with regard to whether chimps copy if dissatisfied or explore if dissatisfied. If you didn’t intend to test these predictions, I would exclude the discussion of them from the introduction.

L120: given your study aim, what were your predictions?

L129: explain how the different life histories might impact their problem solving abilities.

L133: explain what the favorable affordances were and how they differed from scaffolding.

L137: of the 18 individuals, how many were from which subgroup?

L148: were chimps tested individually? It looks like it was a mixture of individually and with others present. If others were present when an individual was being tested, did you account for their opportunity to learn about the task socially?

L159: transparent or opaque tubes?

L161 and 162: inner or outer diameter?

L172: how much previous experience did the chimps have with the different tool options? If some tools were novel, did you check to see if they were afraid of them? It sounds like some individuals had been previously rewarded for using sticks. It would be good to mention their reinforcement history with each material and discuss how this might influence their tool preferences in the tasks.

L175: was the juice reachable using their hands/lips/tongue at a depth of 7cm? Which of the tools provided were functional in stage 1?

L176: was the “bait” the juice? How many attempts could occur during each session or did the session end after the juice was reached the first time? Were the number of different tool choices/attempts per session accounted for in analyses?

L177: how did you determine the end of a session? Why is it important that the tube was emptied of liquid at the end of each session? Did the chimp observe the emptying?

L180: depth 7cm again?

L185: explain which tools were functional and which were not.

L189: I’m not sure how the number four was arrived at for the inserted stick sessions.

L195: why were chimps allowed access to the empty tube and tools between sessions? Did you record whether they attempted to use any of the tools on the tube between sessions? If they did, in the absence of a reward, they may have learned not to prefer certain tools.

L209: kudos for citing R packages!

L214: could the technique also be “not effective”?

L227: did all 18 individuals observe another interacting with the task?

L237: novel to them in terms of this was the first time they had used the technique, but not novel to them because they had seen another use this technique? It is important to separate when individuals acquire information versus when they use information if you are testing something about innovation. I would think innovation is when an individual invents a new solution to a problem without having seen others perform that solution.

L243: what does “potential across phases” mean?

L244: since the functional techniques differed according to phase, effectiveness should be classified only with regard to the phase that the response variable was recorded in. So a hand response in Phase 2 would be Not Effective, whereas it would be classified as Effective in Phase 1. This will be really important for determining whether individuals changed their behavior according to the phase. If this wasn’t already accounted for in the model (it looks like it wasn’t), I would recode the response variable for effectiveness (yes/no) by phase and tool type and rerun the analyses. In the case of the stick, you could break them down into two categories: long (functional only in phase 1) and short (functional in both phases?). Categorizing the response variable in this way will allow you to determine whether individuals changed their behavior according to the phase. The within-individual behavior change will be the key to determining whether individuals exhibit flexibility.

L268: because of what I brought up in the previous comment, it’s not clear whether individuals are rarely choosing Always Effective tools in phase 1 because they are effectively getting the juice using the Partially Effective tools, which is why they then increase their use of Always Effective tools in phase 2. I think categorizing tools as either effective or not according to which phase they are in will be key for understanding the results.

L284: why were only 4 individuals in this phase? How were these individuals chosen?

L286: if Frek and Kindia didn’t retrieve the reward even though it just needed to be licked off of the leaves they pulled out of the tube, were they motivated to participate in the task? It appears that there was no food deprivation prior to testing to increase their motivation in the tasks. How do you know that juice is a high enough reward to motivate them to try hard in tests?

L287: which individuals are “the four individuals who encountered the leafy stick solution first-hand”? It’s not clear whether these are Kindia, Edith, Frek and Pearl or not.

L308: what reward was Manrique et al (2013) using in their tests and was there food deprivation prior to testing? Could this help explain the differing chimp results on a problem solving task? Or perhaps the multi-access box is easier to solve than the test you gave?

L327: in at least a few individuals? At the species level? Specify at what level you think flexibility is occurring.

L331: which study lacked social information, the Davis et al study?

L332: if I understand phases 1 and 2 correctly, they had the opportunity to socially learn about the functionality of the tools by watching conspecifics. These individuals would serve as “demonstrators”. I’m having trouble following “Had the chimpanzees in this study failed to discover ‘Always effective’ techniques, this lack of social information would be a plausible explanation for the relatively diminished behavioural flexibility observed”. Your chimps did have access to social information about effective techniques and they discovered many effective techniques.

L393: explain a bit more about “cultural biases” - the preexisting behaviors in the group, which vary among groups, will be the ones more often used?

L399: explain what “favourable affordances” are.

L483: what factors would you recommend exploring?

Table 1 caption: mention that all individuals observed others attempting/solving the test using a variety of techniques (if this is true).

Table 2: break “Effective in all phases” into 2 columns, one for each phase so readers can see which techniques were effective or not in each phase.

Table 2 caption: note at what hour phase 2 started.

Table 4: excellent that you provided the full model results! Could you note which effects were fixed and which were random? If random effects aren’t included in the table, please add them.

Edinburgh_Dipping_data.xlsx: rows 330, 602, 806, 910, 911, 972, 1106, 1225, 1271, 1496, 1497, 1552, 1553, 1671, 1672, 1804, 1805, 1836, 1837, 2058, 2059, 2123, 2239, 2491, 2769, 2770, 2799, 2819, 2892, 2922, 2936, 2962, 2976, etc are empty. Were they excluded from the analyses? If they were, they should be deleted from the data sheet to avoid confusion. Were the Unknowns excluded from the analyses?

Supplemental Article S1:
- Fig S1: y-axis should say “Always effective” I think.
- Table S1: add random effects to the table. Indicate in the table that Phase refers to Narrow condition and that the intercept includes the Wide condition.


I hope these comments have been useful!

All my best,
Corina Logan
University of Cambridge

References
Hunt, G. R., & Gray, R. D. (2003). Diversification and cumulative evolution in New Caledonian crow tool manufacture. Proceedings of the Royal Society of London B: Biological Sciences, 270(1517), 867-874.

Reviewer 3 ·

Basic reporting

This study by Harrison and Whiten investigates behavioural flexibility in chimpanzees by providing them with different apparatuses that afford different solutions. In the first experimental phase (“Wide Tube”) the juice contained in an apparatus tube could be accessed by using either both bare hands alone or else by recourse to a wide assortment of materials available beside the apparatus. In a second phase (“Narrow Tube”) the opening of the tube was made narrow, thereby rendering inefficient some of the tools and impeding previous solutions. In a third phase, an attempt was made to investigate the structure of their behaviour by observing whether the four chimpanzees might recognize the appropriateness, for accessing the juice, of a suitable tool placed inside the tube, thereby affording a clue that could guide their choice in future.

The results showed a moderate amount of behavioural flexibility because even when chimpanzees’ use of an invariably effective technique increased by 50-fold in the ‘Narrow Tube’ phase, as compared to the previous “Wide Tube” one, the apes were reluctant to relinquish previously-employed albeit now inefficient strategies. Nevertheless, in the “Wide Tube” phase those strategies still comprised 50% of all individuals’ attempts and “no individual that made more than one attempt used “Always effective” techniques for a majority of their attempts”. The “scaffolding” provided by the experimental design did not seem to exert any significant change on behaviour. The high percentage of “Partially effective” attempts in the new “Wide Tube” phase owes to the prepotence that those behaviours rewarded hitherto had acquired in the first “Narrow Tube” phase.

Both the Introduction and Discussion need refinement and clarification. The authors talk of cumulative culture, behavioural flexibility, individual innovativeness, copying of behaviours by conspecifics, etc. It is understood that cumulative culture can emerge only when some individuals are able to innovate solutions that spread horizontally and are passed from one generation to the next. All the topics included in the Introduction and Discussion are appropriate and relevant, yet in the interest of clarity it would be helpful to organize the information in separate sections or paragraphs: for instance, clustering studies that investigate individual innovativeness in order to differentiate them from those that address social learning of newly-acquired behaviours. Also, it would be convenient to separate clearly those studies in which the changes of the apparatus task wholly prevented subjects from using responses rewarded previously, from those which did not prevent them from using old albeit less efficient techniques. In contrast to previous reports, the flexibility observed here could be simply because the old strategies had become so inefficient as to provide no reward at all (cf. Manrique and Call, 2015). Literature on innovation in chimps seems to indicate that they are reluctant to abandon a previously-rewarded response in so far as it had seemed to work: flexibility may be just what we should expect to see whenever a former response was found later to be utterly fruitless.

The authors discuss the results by considering the effects of subgroup yet fail to report how factors such as age (see Manrique and Call, 2015) or hierarchy (Kendal et al., 2015 ) may influence or prevent the acquisition and/or copying of new solutions. In Manrique and Call (2015) it was shown that youngest and oldest individuals were extremely perseverative and failed to abandon old and inefficient responses once the reward contingencies were reversed. Also, another study by the same authors (Manrique and Call, 2011) showed that the oldest orangutans needed considerably more time than younger ones in order to change from the less efficient technique of dipping technique to more efficient sucking in order to drink from a juice container. Interestingly, the only chimpanzee who discovered the most efficient, straw-like technique was a juvenile male. Therefore, it would be interesting for the authors also to analyse their data by focusing on whether age influences behavioural flexibility. Furthermore, because Hopper and colleagues reported a bias towards copying by higher- ranking hierarchy members when individuals were tested in group, it is also necessary to take into account the degree of behavioural flexibility in terms of an individual’s rank: were lower ranking individuals to be those likeliest to change responses, their “innovation” could go unnoticed by conspecifics more readily than were higher-ranking individuals to hit upon the more efficient solution. In lines 328 to 339 the authors refer to the role of an expert model as a constraining factor, which reinforces the need to take into account the hierarchical structures of the two subgroups and the possibility of preferential access by higher-ranking individuals to the apparatus and/or their willingness to change behaviours.


Specific comments:
Lines 45-46. I am not sure that there is evidence of cumulative culture in chimpanzees at all, yet the authors’ statement implies that there is, albeit limited. Given the rarity of the phenomenon it would be convenient to explain briefly what this evidence is.

In line 55 “such as reversal learning paradigms” the work by Manrique and Call (2015) should be added as they also employed a reversal learning paradigm to measure cognitive flexibility as a function of age.

References used:
Kendal, R., Hopper, L. M., Whiten, A., Brosnan, S. F., Lambeth, S. P., Schapiro, S. J., & Hoppitt, W. (2015). Chimpanzees copy dominant and knowledgeable individuals: implications for cultural diversity. Evolution and Human Behavior, 36(1), 65-72. DOI: 10.1016/j.evolhumbehav.2014.09.002
Manrique, H.M., Call, J. (2011). Spontaneous use of tools as straws in great apes. Animal Cognition 14, 213–226.

Experimental design

The main weakness of the study stems from the design: specifically from the fact that individuals were tested in group. Were the authors to have been investigating social learning and transmission of a newly- acquired behaviour, then group testing would have been a mandatory requirement. However, because the stated aim was “to investigate the capability of chimpanzees to alter their behaviour in response to an artificial foraging task” and the method chosen was to test the flexibility of the behaviour and/or innovativeness of the individuals, it might have been preferable to test individuals in isolation and to avoid confusing individual behavioural flexibility with social learning. A more appropriate methodological approach would be first to investigate behavioural flexibility of individuals while tested in isolation with administration of the three apparatus “phases”, and then to select successful individuals and introduce them to the group in order to find out whether or not conspecifics choose the new behaviour after watching how success can be achieved.

When presenting new tasks to a group, factors such as rank, social dynamics, etc., may affect an individual’s attempts to solve problems and thus confound interpretation of the results, particularly if comparisons are to be drawn with previous studies undertaken with isolated individuals. Nor yet is generalization of findings helped by using two different subgroups of chimps with different life histories.

Many factors in the study (social testing, previous history, etc.) could have impinged on the observed behaviour, thereby rendering it difficult to tease apart the specific influence that each of them could exert on the solving abilities of the apes. Now (Given) that the final results of the study have been arrived at, thereby precluding any modification of the experimental procedure, the authors should go the extra mile by discussing how the variable factors mentioned above might impinge on the purported results.

Validity of the findings

Statistical analyses and appropriateness of the interpretation of the results:
The statistical analyses employed are suitable and the conclusions reached are derived appropriately from the data provided. I see no problems there. Notwithstanding the fact that results would have been clearer were subjects to be tested alone.

---

## Round 0.2 · Minor Revisions

Given the fairly substantial amount of suggestions given by the reviewers in the previous round, I obtained a new review of your revision from one of the previous reviewers. I share the reviewer's opinion that the revision has done an admirable job addressing the reviewers' concerns. However, the reviewer makes a number of suggestions that should be considered in another minor revision before I can formally accept your paper. I have a few minor comments of my own:

On lines 139, 140, 246, 517, 610, please change "which" to "that."
Please surround "which are preferred" with commas on line 150.
Retain "which" on line 157.
Comma after studies on line 169.
Different is misspelled on line 174.
Place a comma before which on line 193.
Revise line 200 to read "tasks that they were presented with."
I feel that the paragraph ending on line 211 could benefit from a summary sentence.
Line 212, change "social" to "socially."
I think the intro could still be more focused, setting the stage broadly that there is some controversy over the extent to which chimpanzees might show behavioral flexibility with the work on cumulative culture taking a more tangential role, since it is not the focus of the current study.
Please check that references and spelling are consistent with PeerJ style guidelines.
Place a ., after et al. on line 244.
Please place a ; before however on lines 269 and 358. Place commas after however.
Place a comma after occasions on line 360.
Why was it not possible to document attempts between sessions? Could behaviors during this time not be filmed?
Did the second coder listen to the narration while coding? Was it possible to independently record the behaviors without being influenced by the researcher’s narration? I understand if this was challenging due to the necessity to identify individuals but some note explaining the independence of observations should be made here.
On line 423, ie should be i.e., Similarly, eg. On line 583 should be e.g.,
Switch the order of “only observed” on line 498 to “observed only”
Change “while” to “Although” on line 591 and 674 (or use Whereas).
Move the ‘ in others’ to the end on line 626.
You could avoid repeating the information about sample size limits on examining group differences on lines 697-703 but rather, incorporate the preceding discussion into the limitations section in a more concise way.
Can you show changes in techniques over time, such that there might have been an increased use of the always effective technique throughout the subsequent phases?
Lines 721-723 are also redundant with preceding sections. I don’t think it is necessary to have an extended conclusion following an extended limitations section. Please try to synthesize and make the entire discussion more concise in your revision.

·

Basic reporting

No comment.

Experimental design

Dear authors,
You did an excellent job of clarifying terms and concepts in your revision. Your treatment of testing in a social context and how this relates to behavioral flexibility was also well done and provides a nice base for future studies to build on. Additionally, because you now provide information on the effectiveness of each technique, I understand how you treated individual effects in your models. Since there was very little individual variation in your study, I can see why it makes sense to discuss the results in the context of the whole group rather than at the level of the individual.

Validity of the findings

No comment.

Comments for the author

A few additional thoughts/comments came up as I read your revised manuscript (line numbers are from the tracked changes Word document):

100 - “cumulative progressions rather than unconnected innovations”.

Couldn’t cumulative culture result from both cumulative progressions and unconnected innovations? The assumption would be that only those unconnected innovations that were more functional would end up being incorporated into the repertoire.

110 - “Social learning, innovation and behavioural flexibility are expected to work in concert to support cumulative culture – an individual innovates an improvement to a behaviour or tradition, and this improvement is passed on via social learning to other group members.”

It’s not clear where flexibility is in this example. It seems from later sentences in this paragraph that flexibility involves the ability to choose the more functional behavior, which keeps culture adaptive?

165 - the western scrub-jay is now called the California scrub-jay…I’m not sure if you should update the name in your paper even though you are referring to the name used in the paper you discuss, but I thought I would mention it.

257 - didn’t the orangutans abandon some solutions and solve some, but not all, of the new solutions in the Manrique study? If so, I would rephrase the sentence to indicate the orangutans didn’t completely fail this task, they just performed more poorly relative to the other species.

320 - “scaffolding in this manner”
Please clarify what “in this manner” means because I don’t quite know, which makes it unclear how it differs from learning through exposure to artefacts.

480 - “Note that all techniques (see Table 2) were effective in the ‘Wide Tube’ phase, in which there were few constraints upon potential solutions.”

It sounds like there were no constraints in the Wide Tube phase. If there were “few constraints” list what they were.

758 - I would change “cognitive flexibility” to “behavioural flexibility” because the latter is the term used throughout the paper and it is the behavior that is measured.

761 - “and so the age of many individuals in the current study may have impaired their performance”

Because most of the individuals in your study were over 27 years of age?


All my best,
Corina Logan
University of Cambridge

---

## Round 0.3 · Minor Revisions

Thank you very much for your careful response to the last round of review. I will be happy to accept this manuscript as soon as you attend to the following very minor edits:

Line numbers below refer to the numbering in the PDF MS to be reviewed (without tracked changes).

Please avoid double parentheses (lines 66, 103-104, 598, replace second ( with;

On lines 76, 106, 134, 222, 573, replace “which” with “that”

Punctuation is missing on line 81. However should begin a new sentence and be followed with a ,

References on lines 99-100 are in neither alphabetical or chronological order. Same issue on line 136 and 487. Please use a consistent system. If referring to the order of species listed, place each ref after each species instead. Please do the same on lines 565 rather than repeating species names. Insert a space between “specieswere” on line 184. Place a , after “social learning)”

Place a comma after however on line 330.

It might be important to indicate why you describe the lab environment as more restrictive compared to the zoo environment (lines 232-240). It is not the case that all lab chimpanzees live in restricted environments, nor that all zoo chimpanzees do not. Some lab environments may be highly enriching and provide chimpanzees with greater experience with human artifacts etc. Some further exposition here would be wise. Could you delete lines 671-673, since you’ve already discussed this issue earlier in the discussion.

There is a blank page after line 251 and a large blank space after line 420.

---

## Round 0.4 · accepted · Accept

Thank you so much for your careful attention to these final corrections. And thank you for submitting such interesting work to PeerJ.